# Modeling and Experiments of Droplet Evaporation with Micro or Nano Particles in Coffee Ring or Coffee Splat

**DOI:** 10.3390/nano13101609

**Published:** 2023-05-11

**Authors:** Hongbing Xiong, Qichao Wang, Lujie Yuan, Junkai Liang, Jianzhong Lin

**Affiliations:** 1State Key Laboratory of Fluid Power and Mechatronic Systems, Zhejiang University, Hangzhou 310027, China0020586@zju.edu.cn (J.L.); 2Zhejiang Provincial Engineering Research Center for the Safety of Pressure Vessel and Pipeline, Ningbo University, Ningbo 315211, China

**Keywords:** droplet evaporation, coffee ring effect, particle deposition, phase field, diffuse interface model, additive manufacturing

## Abstract

Experimental and numerical experiments were carried out to study the coffee rings or coffee splats formed by droplet evaporation with micro or nano polystyrene sphere particles (*D_p_* = 10 μm or 100 nm). Particle image velocimetry (PIV) and a high-resolution camera were used in this experiment, along with a temperature-controlled heater and a data-acquisition computer. The results showed that a nano particle could form a homogeneous coffee splat, instead of the common coffee ring formed when using micro particles. In order to account for this phenomenon, this paper developed a complex multiphase model, one which included the smooth particle hydrodynamics (SPH) fluid model coupled with the van der Waals equation of state for droplet evaporation, the rigid particle model of finite-size micro particles, and the point–particle model of the nanometer particles. The numerical simulation was operated on a GPU-based algorithm and tested by four validation cases. A GPU could calculate 533 times the speed of a single-core CPU for about 300,000 particles. The results showed that, for rigid solid particles, the forms emerged spontaneously on the wall, and their structure was mainly affected by the boundary wettability, and less affected by the fluid flow and thermal condition. When the wall temperature was low, it was easier for the particles to be deposited on the contact line. At high wall temperature, the coffee ring effect would be weakened, and the particles were more likely to be deposited in the droplet center. The hydrophilic surface produced a larger coffee ring compared to the hydrophobic surface. The experimental and numerical results proved that particle size could play a significant role during the particle deposition, which may be a possible route for producing uniform-distributed and nano-structure coatings.

## 1. Introduction

After evaporation of liquid drops containing particles, there might be an even or uneven distribution pattern in the remaining particles. The most typical one is the “coffee ring effect” [1], as when a drop of coffee drops on a table and evaporates completely, leaving a stain on the surface, with particles more concentrated around the contact line. In recent years, coffee rings have found wide applications in many industries, such as analytical chemistry, environmental pollution monitoring, low-cost biomolecular identification, forensic analysis and microstructural printing [2,3,4,5].

The coffee ring effect is affected by many factors, such as wall temperature [6], wall hydrophilicity [7], wall micro-structure [8], contact line movement speed [9], particle motion [10], particle shape [11], particle concentration [12], droplet size [13] and so on. Patil et al., showed that the morphology of the coffee ring is related to the boundary temperature, contact angle and particle concentration [12]. He and Derby, based on their experimental results, developed a mathematical model for describing the transition from coffee ring to uniform deposition pattern [13]. Although there have been many experiments in this area, it was noticed that there has been no clear discussion on the effect of particle size on the coffee ring formation, which will be the main topic discussed in this paper. Additionally, previous experiments have found that droplet evaporation with nano particles might deposit a uniform stain where particles distribute themselves evenly within the contact line. This article calls it “coffee splat” to distinguish it from the common coffee ring.

In order to describe the coffee ring effect, it is necessary to find an accurate numerical model. Many numerical algorithms have been developed to discuss the coffee ring effect. These methods include particle-based algorithms [14], the hydrodynamics algorithm of the Euler mesh method [15], the lattice Boltzmann method (LBM) [16] and so on. Zhao [16] used the LBM method to discuss the influence of interfacial displacement in the coffee ring effect. Xu [17] used a discrete element model to describe particle packing. Whereas other studies of coffee rings focus on homogeneous particles, this article has considered the simultaneous movement of two particles of different sizes. This article uses the smoothed particle hydrodynamics (SPH) method and the van der Waals equation of state to simulate the phase change of droplet evaporation on a superheated wall. The motion of the containing micro or nano particles are described by rigid particle or point–particle models.

This paper discusses particle motion when a droplet with differently-sized particles evaporates on a superheated wall at different temperatures. Firstly, this article introduces the experimental platform in Section 2.1 and the results in Section 2.2. Section 3 then introduces the numerical model, and Section 4 shows the verification of numerical model. Numerical results are presented in Section 5, discussing the effects of particle size difference (Section 5.1 and Section 5.2), substrate temperature and particle concentration (Section 5.3), and substrate wettability (Section 5.4) on the evaporation dynamics and deposition pattern.

## 2. Experimental Methods

In this experiment, a single droplet was released from the injection of a pipette onto a heated wall with a constant temperature, and its evaporation process was recorded with two cameras. Several cases of experiments of droplet evaporation, both without particles and with micro and nano particles, were carried out.

### 2.1. Setup and Procedure

During the experiment, the ambient temperature was 13 °C, atmospheric pressure was 102 kPa, and the relative humidity of the air was 74%. This experiment employed three sizes of polystyrene sphere particles, as *D_p_* = 1000 μm, 10 μm and 100 nm, respectively. The particles’ mass fraction *w* was from 0.025 wt% to 2.5 wt%. Firstly, it was necessary to prepare the dispersion of the particles in the deionized water. In order to obtain particles with good dispersion, the ultrasonic generator was used to break the particle clusters and prevent the agglomeration of particles [18].

These experiments were carried out on an in-house built platform. Figure 1 gives the photo and schematic diagram for the experimental setup. The experiments were performed in seven steps: (1) Prepare the dispersion of polystyrene particles and use the ultrasonic generator to prevent the particles’ agglomeration. (2) Use isopropyl alcohol and deionized water to wipe the steel plate (100 × 100 × 0.3 mm^3^) and apply silicone grease between the heating table and the substrate to ensure that the wall surface is clean and the temperature distribution of the substrate is uniform during heating. (3) Use two rotary tables to calibrate the heating table to level. Set an appropriate heating source, depending on the wall temperature and ambient temperature. Turn on the heating table to preheat the wall to the specified temperature. Use the thermocouple to monitor wall temperature. (4) Extract the dispersion fluid with a micro-pipette and install it on the stepper motor to ensure a uniform injection speed without bubbles. Inject a quantitative amount of dispersion (10.0–25.0 μL) onto the wall. (5) Use camera 1 to help to adjust the laser strafing ranges. This allows the laser to illuminate the center of the droplet. If space allows, the operator can use optical lenses such as spectroscope and prism to adjust laser width. (6) Use camera 2 to observe the outline of the droplet and the particles’ motion. Fine-tune the position of the laser to help ensure that the imaging is clearly visible. (7) Accomplish data post-processing analysis.

### 2.2. Experimental Results

Figure 2 shows the typical coffee ring patterns that formed at different combinations of micro (*D_p_* = 10 μm) and nano particle (*D_p_* = 100 nm) mass fraction at a given wall temperature of 40 °C. The mass fraction of the particle ranged from 0 to 0.025 wt%. For the first case, containing only micro particles, the coffee ring effect is the clearest. A white, sharp ring was left on the droplet contact line, while the residual part of the droplet footprint was randomly either white or dark. This meant that particles inside the contact line were distributed unevenly, and most of the area was full of innumerable coarse grains. For the second case with half micro particles and half nano particles, the ring had a larger width and the particles were more evenly distributed and packed adjacent to the contact line. Only two or three coarse grains have been found. For the third case, containing only nano particles, the ring had the largest width and most even distribution among these three cases. No coarse grains were found. This means that the particles’ size has a major effect on the particle deposition pattern of droplet evaporation.

The unfavorable stains of the coffee ring effect might be reduced by adjusting particle size and other conditions. Furthermore, this article wants to find out whether the residue could have a uniform coating if more nano particles were deposited. Thus, this article picked only the nano-sized particles of 100 nm and enlarged their concentration up to 2.5 wt%, and then evaporated the droplet on the wall at different temperatures. Figure 3 shows the experimental results of particle deposition patterns. Compared to the results in Figure 2, the cases in Figure 3 resulted in wider thicknesses of coffee ring, since more particles were deposited on the contact line, resulting in a higher particle concentration. The inner part of the droplet footprint was different for different wall temperatures. At low temperatures, the coffee ring effect is clear again even with only nano particles, while at higher temperatures, the coffee ring effect gradually weakens. At a wall temperature of 80 °C, the particle deposition pattern is even and uniform, which is called “coffee splat”.

Both the particle size and the wall temperature affected the ring or splat diameter. Figure 4 shows the diameter of the remaining coffee ring or coffee splat after the experiments in Figure 2 and Figure 3. Both increasing the concentration of nanoparticles and increasing the temperature reduced the ring/splat diameter. Increasing the nano particle concentration from 0 to 2.5 wt% dropped the ring diameter 4.8%. Increasing the wall temperature from 30 °C to 80 °C dropped the ring diameter 13%.

Due to the high concentration of the particles, the whole drop appears opaque, and PIV technology cannot be used to obtain the flow structure inside the droplet. In order to further examine the motion of particles, this paper adopted the SPH method to simulate the motion of particles in the droplet and examine the formation mechanism of coffee ring and coffee splat.

## 3. Numerical Simulation Model

### 3.1. Governing Equations

This section introduces the fluid governing equations and the numerical discretization scheme. SPH is a method originally proposed by Lucy [19] and Gingold and Monaghan [20]. This article adopts this numerical model to simulate the multiphase problem, which is based on SPH and the diffuse interface method [21].

The fluid governing equations are given in Lagrange form. The time derivative of the SPH particles’ density, velocity and internal energy can be rewritten as the summation of surrounding particles [22]. For example, particle density can be obtained by counting the numerical density of local particles. Equations (1)–(3) are the density statistical equation, momentum conservation equation, and energy conservation equation, respectively [23]. The mass density and the derivative of velocity and energy density with time of a fluid particle *a* is calculated by the following formula:(1)$$\rho_{a}=\sum_{b}m_{b}W_{ab}\left({\vec{x}}_{a}-{\vec{x}}_{b}/h_{ab}\right)$$
(2)$$\begin{array}{ll} \frac{d{\vec{v}}_{a}}{dt}= & \sum_{b}m_{b}\left(\frac{M_{a}}{\rho_{a}^{2}}+\frac{M_{b}}{\rho_{b}^{2}}\right)\cdot \nabla_{a}W_{ab}+\\ & \sum_{b}m_{b}\left(\frac{M_{a}^{H}}{\rho_{a}^{2}}+\frac{M_{b}^{H}}{\rho_{b}^{2}}\right)\cdot \nabla_{a}W_{ab}^{H}+\\ & {\vec{F}}_{Solid-Fluid}+\vec{g} \end{array}$$
(3)$$\begin{array}{ll} \frac{du_{a}}{dt}= & \frac{1}{2}\sum_{b}m_{b}\left(\frac{M_{a}}{\rho_{a}^{2}}+\frac{M_{b}}{\rho_{b}^{2}}\right):\left({\vec{v}}_{a}-{\vec{v}}_{b}\right)\nabla_{a}W_{ab}+\\ & \frac{1}{2}\sum_{b}m_{b}\left(\frac{M_{a}^{H}}{\rho_{a}^{2}}+\frac{M_{b}^{H}}{\rho_{b}^{2}}\right):\left({\vec{v}}_{a}-{\vec{v}}_{b}\right)\nabla_{a}W_{ab}^{H}+\\ & \sum_{b}\frac{2\kappa_{ab}m_{b}\left({\vec{x}}_{a}-{\vec{x}}_{b}\right)\cdot \nabla_{a}W_{ab}}{\rho_{a}\rho_{b}\left(r_{ab}{}^{2}+0.01h^{2}\right)}\left(T_{a}-T_{b}\right) \end{array}$$
where $\rho_{a}$ (kg/m^3^) is the fluid density of particle *a*, *m_a_* is mass of particle *a*, $W_{ab}\left({\vec{x}}_{a}-{\vec{x}}_{b}/h_{ab}\right)$ (m^−3^) is the smooth function, ${\vec{x}}_{a}$ (*m*) is the coordinate of particle *a*, $h_{ab}=(h_{a}+h_{b})/2$ (*m*) is the average smooth length of particles *a* and *b*. Additionally, ${\vec{v}}_{a}$(m/s) is the fluid velocity of particle *a*, ***M****_a_ *(*Pa*) is the stress tensor for particle *a,* $\nabla_{a}W_{ab}$ (*m*^−4^) is the gradient of the smooth function, ${\vec{F}}_{Solid-Fluid}$ (*N*) represents the force of the solid particles acting on the fluid particles (its formula will be given in Section 3.2) and $\vec{g}$ (m/s^2^) is the gravity acceleration. Additionally, *u_a_* (J/kg) is the internal energy density of particle *a*, *T_a_* (*K*) is the temperature of particle *a*, and $\kappa_{ab}=2\kappa_{a}\kappa_{b}/\left(\kappa_{a}+\kappa_{b}\right)$ (W/m/K) is the harmonic mean of the thermal conductivity of particles *a* and *b*.

The stress tensor ***M*** contains a pressure term, a viscous term and a surface tension term:(4)$$M=-pI+\eta_{S}\left(\nabla \vec{v}+\nabla {\vec{v}}^{T}\right)+M_{C}$$
where *p* (*Pa*) is the pressure, $\eta_{S}$, (*Pa·s*) is shear viscosity and ***M_C_*** (*Pa*) is the Korteweg tensor. This tensor is expressed in terms of the density spatial gradients and is operating mainly at the vapor-liquid interface,
(5)$$M_{C}=K\left(\rho \nabla^{2}\rho +\frac{1}{2}{\left|\nabla \rho \right|}^{2}\right)I-K\nabla \rho \nabla \rho$$
where *K* (m^7^/s^2^/kg) is the gradient energy coefficient. This term vanishes when the fluid temperature reaches critical temperature. In the simulation, the stress tensor ***M*** is divided into two parts [24,25],
(6)$$M=\left[-\left(\frac{\rho {\overset{-}{k}}_{b}T}{1-\overset{-}{\beta }\rho }\right)I+\eta_{S}\left(\nabla \vec{v}+\nabla {\vec{v}}^{T}\right)\right]+M^{H}$$
where $M^{H}$ is long-range interaction term and $M^{H}=M_{C}+\left(\overset{-}{\alpha }\rho^{2}\right)I$ [21,26].

To close the governing equations, this paper uses the van der Waals equations of state to describe the relation between pressure, density, internal energy and temperature,
(7)$$p=\frac{\rho {\overset{-}{k}}_{b}T}{1-\overset{-}{\beta }\rho }-\overset{-}{\alpha }\rho^{2}$$
(8)$$u=c_{v}T-\overset{-}{\alpha }\rho$$
where ${\overset{-}{k}}_{b}$ (m^2^/s^2^/K) is ideal gas constant, $\overset{-}{\beta }$ (m^3^/kg) is the parameter of repulsive force, $\overset{-}{\alpha }$ (m^5^/s^2^/kg)is related to the attractive force and $c_{v}$ (J/kg) is the specific heat capacity.

In this article, the following smooth function is used [24]:(9)$$W\left(q\right)=\alpha \{\begin{array}{ccc} q^{3}-6q+6 & & 0\leq q<1\\ {\left(2-q\right)}^{2} & & 1\leq q<2\\ 0 & & otherwise \end{array}$$
where $q=\left|{\vec{x}}_{b}-{\vec{x}}_{a}\right|/h_{ab}$, $h_{a}=\varepsilon_{k}{\left(m_{a}/\rho_{a}\right)}^{-dim}$ (*m*) is smooth length, and $\varepsilon_{k}=1.8$ is adopted in this paper. Nonetheless, α is 1/(7h), $1/\left(3\pi h^{2}\right)$ and $15/\left(62\pi h^{3}\right)$ for one-, two- and three-dimensional problems, respectively.

### 3.2. Particle Modeling

There are two kinds of particle models adopted in this paper, which are rigid particles that can interact with the flow field and point–particles only affected by the drag force and gravity. The former is used for micro particles, while the latter is used for nano particles.

Rigid particle velocity can be divided into two components, translational velocity and rotational velocity. The governing equation of rigid particle action is as follows [27]:(10)$$\frac{d\vec{X}}{dt}=\vec{V}$$
(11)$$\frac{D\vec{V}}{Dt}=\frac{{\vec{F}}_{Fluid-Solid}}{M}+\vec{g}$$
(12)$$\frac{D\vec{\Omega }}{Dt}=\frac{{\vec{T}}_{Fluid-Solid}}{I}$$
where *M*, *I*, $\vec{V}$ and $\vec{\Omega }$ denote the particle mass, particle inertia about the centroid ***C***, the translational velocity and rotational velocity, respectively. Based on the balance equation of fluid and particle, for SPH modeling, the interaction force ${\vec{F}}_{Fluid-Solid}$ (*N*) and torque ${\vec{T}}_{Fluid-Solid}$ (*N*·*m*) are as follows:(13)$${\vec{F}}_{Fluid-Solid}=\sum_{a\in Solid}\sum_{b\in Fluid}m_{b}\left(\frac{M_{a}}{\rho_{a}^{2}}+\frac{M_{b}}{\rho_{b}^{2}}\right)\cdot \nabla_{a}W_{ab}$$
(14)$${\vec{T}}_{Fluid-Solid}=\sum_{a\in Solid}\sum_{b\in Fluid}\left[-\left(\frac{{\vec{x}}_{a}+{\vec{x}}_{b}}{2}-{\vec{x}}_{C}\right)\times m_{b}\left(\frac{M_{a}}{\rho_{a}^{2}}+\frac{M_{b}}{\rho_{b}^{2}}\right)\cdot \nabla_{a}W_{ab}\right]$$

According to theoretical mechanics, the velocity ${\vec{v}}_{a}$ (m/s) and acceleration ${\vec{a}}_{a}$ (m/s^2^) of each child particle, consistent with that of rigid particles, can be written as follows:(15)$${\vec{v}}_{a}=\vec{V}+\vec{\Omega }\times \left({\vec{x}}_{a}-{\vec{x}}_{C}\right)$$
(16)$${\vec{a}}_{a}=\frac{d\vec{V}}{dt}+\frac{d\vec{\Omega }}{dt}\times \left({\vec{x}}_{a}-{\vec{x}}_{C}\right)+\vec{\Omega }\times \left({\vec{v}}_{a}-\vec{V}\right)$$
where the italicized baseline *a* represents the SPH particles that make up the rigid body particles.

However, when dealing with the actual flow problem, people often encounter the problem of multi-scale particles. When the particle’s size is much smaller than that of the droplet, the effect of the particle on the flow field can be ignored [28]. The Lagrangian approach is used to trace the particles individually. Particle acceleration is tracked after release by considering the drag force of the heated fluid and the gravity. The driving forces on a particle include the pressure gradient force, the viscous drag force and the gravitational force. When a particle’s size is smaller than 10 μm, the drag force becomes the primary one in the transfer of momentum. The governing equation of the point–particle can be described by:(17)$${\vec{F}}_{p}=m_{p}\frac{d{\vec{V}}_{p}}{dt}$$
(18)$$\frac{d{\vec{V}}_{p}}{dt}=\frac{3}{8}\frac{\overset{-}{\rho }}{\rho_{p}}\frac{C_{D}}{r_{p}}\left|\vec{V}-{\vec{V}}_{p}\right|\left(\vec{V}-{\vec{V}}_{p}\right)$$

In our case, the drag coefficient is:(19)CD={24/RepRep<124/Rep⋅(1+0.15 Rep0.687)1<Rep<10000.44Rep>1000

The Reynolds number for particle *p* is defined as:(20)Rep=ρfDp|V→−V→p|μf
where $D_{p}$ (*m*) is particle diameter, and $\rho_{f}$ (kg/m^3^) and $\mu_{f}$ (*Pa*·s) are the local fluid density and viscosity around the particle, respectively.

### 3.3. Numerical Setup

#### 3.3.1. Boundary Condition

In the simulation, the droplet is placed on a heated wall in a rectangular box filled with vapor. Each side of the box length is 7.5 times that of the droplet’s diameter. An open boundary (*z* = 7.5 *D*) is added, as proposed by Tafuni [29] for the computational domain. The particle’s physical properties at the open boundary are obtained from the interpolation of mirror particles inside the flow field. Because the open boundary and the heated wall are far apart, the following numerical cases assume that the pressure, velocity, and temperature gradient at the open boundary are all zero. The vapor particles are able to move freely in and out of the open boundary. This keeps the pressure inside the system at a relatively balanced level. The boundary (*z* = 0) is kept at a constant temperature, with nonslip boundary conditions. The periodic boundary condition is applied along *x* and *y* direction.

#### 3.3.2. Initial Condition

The droplet is placed on the wall, and a certain number of particles are initially randomly distributed inside the droplet, including rigid particles and point–particles. The initial density and temperature of the droplet are $\rho_{l}=1.3875$ (it is a dimensionless variable, the original unit is kg/m^3^) and $T_{l}=0.6$ (it is a dimensionless variable, the original unit is *K*), respectively. The initial state is located on the binodal line, which is in a gas–liquid equilibrium state [30]. The initial density and temperature vapor are $\rho_{v}=0.008\rho_{l}$ and $T_{v}=T_{l}$, respectively.

This section describes the deposition pattern of particles during the evaporation of droplets on superheated walls. The wall superheat degree is 0.1, 0.2 and 0.3, the droplet diameter is 8, the point–particle diameter is 0.25, and the number of point–particles is 8000. There are two types of rigid-body particles, with diameters of 1 and 0.5, and the number of rigid particles is 14.

#### 3.3.3. Parallel Computing Method

In order to save the time consumed by SPH simulation, this paper proposes two CUDA parallel computing strategies. In short, strategy one lets one thread handle the computation of one particle. Strategy two uses the shared memory, where multiple threads process work together to solve the computation of multiple particles. Shared memory is allocated per thread block, so all threads in one block have access to the same shared memory. Strategy two saves the thread time consumed by reading the data. Figure 5 shows the computational speedup ratio (*Sr*) of different parallel strategy algorithms, as compared to CPU single-core computation. The black and red dots correspond to the results of strategy one and strategy two, respectively. The computing machine has an Nvidia GeForce RTX 3090 graphics card and an Intel 12490F CPU. When the number of particles reached 319,693, the calculated speedup ratio (*Sr*) reached 533 times. This shows that the parallel method is suitable for dealing with SPH simulation problems.

#### 3.3.4. Simulation Parameters in SPH Modeling

In order to facilitate the simulation and reduce rounding error, all parameters are dimensionless. Table 1 shows the parameters involved in the simulation process. Table 2 contains key parameters for each section of this article, including boundary superheat degree (Δ*T*) and wettability. Note that, while this simulation system is designed to examine the mechanism, it could not be set as exactly as the experiments; even with the parallel computing method, the simulation’s time consumed is still challenging.

## 4. Numerical Model Validation

In this section, the numerical simulation is verified by a series of examples, which shows that the SPH method can accurately describe the droplet evaporation on the superheated wall. Four validation cases have been conducted to compare with the analytic or experimental results: (1) flow pass of a rigid sphere; (2) sessile droplet evaporation; (3) droplet flow pattern during evaporation; and (4) particle distribution and precipitation pattern after evaporation.

It is worth noting that the SPH method should be validated first for one phase flow (no solid particles) by comparison with results in the existing literature or extant data. For details, please refer to our published paper [31].

### 4.1. Flow Past a Sphere

This subsection simulates the flow of fluid through a stationary sphere with a Reynolds number equal to one and theoretical *C_d_* = 24/*Re*. A total of 10 layers of SPH particles were arranged from inside to outside for this stationary sphere. The particles’ smooth length was *h* = 1.8Δ*x*. The inlet was a constant velocity boundary condition. The outlet was a constant pressure boundary condition. Other boundary conditions were periodic boundary conditions. Figure 6 shows the numerical simulation’s results. The different lines represent different levels of numerical precision. For the GPU parallel computational method, single precision calculation is faster than double precision calculation. Both numerical precisions can achieve their goals. The simulation results are in good agreement with the theoretical results. It can be seen from Figure 6a that the density of the fluid changed to some extent, because this article adopted the van der Waals equation of state for this example, and the flow velocity of the fluid was relatively large, which leads to a certain change in the density of the fluid. The calculation formula of the drag coefficient of the SPH sphere is:(21)$$C_{d}=\frac{\left|{\vec{F}}_{n}\right|}{\frac{\pi }{8}D_{p}^{2}\rho_{f}v_{\infty }^{2}}$$
where, in this paper, particle diameter is $D_{p}=2.5$ (*m*), fluid density is $\rho_{f}=1$ (kg/m^3^) and incoming fluid flowing velocity $v_{\infty }=0.2$ (m/s), ${\vec{F}}_{n}$ (*N*) is the drag force. For the SPH method, the summation force of the boundary particle is dragging force ${\vec{F}}_{n}$, which is similar to Equation (19); the calculation formula is:(22)$${\vec{F}}_{n}=\iint_{S_{p}}\left(-pI+\left[\mu \left(\nabla \vec{v}+\nabla {\vec{v}}^{T}\right)\right]\right)\cdot \vec{n}\;dS=\sum_{a\in Solid}m_{a}\left[\sum_{b\in Fluid}m_{b}\left(\frac{M_{a}}{\rho_{a}^{2}}+\frac{M_{b}}{\rho_{b}^{2}}\right)\cdot \nabla_{a}W_{ab}\right]\cdot \vec{n}$$

### 4.2. Droplet Flow Structure

In order to observe the flow field during the generation of coffee rings, PIV technology was utilized to photograph the flow structure inside the droplet. Polystyrene particles were added (10 μm) as tracer particles inside the droplet. The mass fraction of the polystyrene particles was 0.025 wt%. The movement track of the particles on the surface of the droplet during the evaporation process was recorded. Figure 7 is the flow field inside the droplet in experiment and simulation. The total number of SPH particles was 200 × 200, and the particles’ smooth length *h* = 2.0∆*x*. In the numerical simulation, the bottom of *Y* direction is a constant temperature non-slip wall, the top of Y direction is an open boundary condition with constant steam concentration, and *X* is a periodic boundary condition. For Figure 7b, the vapor SPH particle is hidden. In order to make the image clear, the steam particles are hidden and the fluid particles are post-processed. Since there are many particles in this example, the velocity distribution diagram is drawn by taking 16 nearby particles as a group. The velocities of several particles in each background grid were averaged over the sum into fewer particles for display. The typical velocity inside the droplet flow field is 3.33 mm/sec. The flow structure obtained by calculation is similar to that obtained by experiment. Figure 7c shows the SPH results and the experimental results of the flow velocity at the center of the droplet.

### 4.3. Evaporation of a Sessile Droplet

This subsection describes a droplet evaporation experiment and numerical simulation and compares the simulation result to the experimental result. The droplet initial volume is *V*_0_ = 10 μL, and wall temperature superheat degree is 75 °C. It takes *T_End_* = 120 s for the droplet to completely evaporate. For SPH modeling, the computational domain size is four times the droplet’s diameter. The total number of SPH particles is 100 × 100, and the particle smooth length *h* = 2.0∆*x*. An open boundary is adopted to release the vapor during evaporation. The boundary particles are arranged according to the mirror particles. There are periodic boundary conditions in the *X* direction. For the *Y* direction, there is an open boundary and no slip heated boundary. As shown in Figure 8, the SPH simulation results agree well with the experimental data. This numerical method is reliable for dealing with the droplet evaporation problem.

### 4.4. Particle Distribution Pattern

Figure 9 shows the particles’ pattern in the presence of multiple particles. This subsection gives the distribution of particles at the later stage of droplet evaporation, and where the fluid has almost completely evaporated. The particle diameter of large particles is 1000 μm, and the particle diameter of small particles is 100 nm with mass fraction of 0.025 wt% in volume of 10 μL. The total number of the SPH fluid particle is 80,000. The particle smooth length is *h* = 1.8∆*x*. In the numerical simulation, the bottom of *Z* direction is a constant temperature wall, the top of *Z* direction is an open boundary condition, and the *X* and *Y* directions are periodic boundary conditions. It can be seen that large particles form regular structures after evaporation, and similar results are also obtained in the SPH numerical simulation. The final precipitation pattern obtained by calculation is similar to that obtained by experiment.

## 5. Numerical Results and Discussion

This section discusses some of the factors that can affect droplet evaporation and the coffee ring effect, including boundary superheat degree, boundary wettability, particles size, droplet evaporation and so on.

### 5.1. Effect of Rigid Particles

The numerical simulation results in this section show that rigid particles can also promote the formation of the coffee ring effect. Due to the particle sedimentation, the droplets are affected by the particles and spread out on the wall’s surface.

Figure 10 and Figure 11 show the motion evolution of rigid particles of two sizes during droplet evaporation; the black spheres represent rigid particles. The density of fluid particles can be distinguished by their color. In order to visually represent the motion of liquid particles and vapor particles, vapor particles with a density of less than 0.1 are hidden. It can be seen that the larger-diameter particles always stay inside the droplet, while the smaller-diameter particles follow the steam out of the droplet. For a particle with a diameter of 1.0, when the boundary superheat is low, the droplet unfolds gradually on the wall with the evaporation of the droplet, and its final thickness is approximately the diameter of the rigid particle. For particles with a diameter of 0.5, when the superheat is low, the particles will be carried away from the droplet center along with the steam.

Figure 12 shows the point–particles’ deposition pattern in the absence of rigid particles. The boundary superheat degree is 0.2. When the droplet was just released to the wall, the impact between the droplet and the wall caused some point–particles to settle in the center area of the pattern. With the passage of time, the deposition range of point–particles gradually expanded. In the presence of rigid particles, point–particles are less likely to leave the droplet with evaporation, and the deposition particles form a ring-like pattern at the contact line. In the absence of rigid particles, the point–particles are more dispersed, forming a larger distribution range. This may be because the rigid particles limit the spreading of the droplets, so that the point–particles are more likely to settle at the contact line. The rigid particles make the deposition of the point–particles more uneven.

### 5.2. Effect of Point–Particles

Figure 13 shows the particles’ deposition pattern as point–particles on the wall with superheat of 0.1 in different time periods. It can be seen that the deposition range of particles gradually expands, while the total number of particles deposited in each time period gradually decreases. With the passage of time, the total precipitation of particles at first reached a peak and then rapidly decreased. With the presence of the point–particle, the time node of maximum particle precipitation develops gradually. As can be seen from Figure 13, the precipitation range of particles gradually increases over time. The reason is that the point–particle is very capable of following the fluid’s flow, and thus, it would constantly expand with the evaporated fluid.

### 5.3. Effect of Wall Superheat Degree

When the boundary superheat degree is low (∆*T* = 0.1, 0.2, 0.3), with the gradual evaporation of the droplet, the droplet is spread out on the wall, and its thickness is similar to the diameter of the rigid particle. The arrangement of the rigid particle appears to follow a certain rule. After releasing the droplet onto the wall, though, the droplet exhibited different evaporation modes as the boundary superheat degree increased.

Figure 14 shows the results of the point–particle sinking under different levels of superheat. Figure 14a represents the number of precipitated particles in different time periods. Figure 14b shows the distribution range of precipitated particles in different time periods. Figure 14c represents the average distribution number of precipitated particles along the droplet’s centroid after the droplet evaporates completely. Figure 14d represents the total precipitation amount and average precipitation range of particles under different superheat. Compared with the conditions of 0.2 and 0.3 superheat, when the superheat is 0.1, the number and range of precipitated particles are the largest, and at their maximum during the period from the initial release of droplets to the complete evaporation. It can be seen from Figure 14c that the distribution of particles is somewhat related to the wall superheat. When the wall superheat is 0.1, the precipitation range of particles has a peak value between *R* = 6–7, that is, the coffee ring effect occurs. As can be seen from Figure 14d, with the increase of wall superheat, the total precipitation amount and precipitation range of particles decreased.

Figure 15 shows the particles’ final deposition pattern. The color of the particle represents the time when it settled. Under low superheat conditions, particles are more likely to precipitate near the contact line and the center of the droplet, which forms the structure of coffee ring and “coffee splat”. The final states of point–particles can be classified into two types: (1) leaving with the steam and entering the steam; and (2) migrating to the wall. Due to the non-slip boundary condition and the gravity, eventually, those point–particle will be deposited. As the droplet evaporates, more and more point–particles are deposited at the gas-liquid boundary. Under the condition that the wall keeps to a low degree of superheat, the particles are more likely to settle at the contact line and the center of the droplet, namely, the coffee ring and the coffee eye. With the increase of the wall temperature, more particles will settle in the center of the droplet, and the scope of particle deposition is gradually reduced.

### 5.4. Effect of Surface Wettability

The results show that the microstructure of the wall affects the precipitation of particles [32]. Reflected to the macroscopic scale, this becomes the wall contact angle. This subsection shows the influence of wall wettability on the coffee ring effect. The boundary superheat degree is kept at 0.1. The above numerical results are conducted on a hydrophilic surface. However, if on a hydrophobic surface, the droplet will spread into a thicker layer of liquid film, which will result in a more uniform precipitate. After evaporation, the particles will be evenly spread on the wall surface, and finally form the pattern of a sheet structure. As shown in Figure 16, on the hydrophobic wall, the particles will agglomerate in the droplet. As the droplet evaporates, a stable structure of the particle stack will finally be formed. As in Section 5.1, the particle distribution on the hydrophilic wall shows the structure of a coffee ring. However, when the wall is hydrophobic, as shown in Figure 17, the deposition particles are more concentrated and less in number. This may be related to the smaller contact area between the droplet and the hydrophobic wall, so the point–particles are more likely to leave with the steam.

## 6. Conclusions

This article has carried out the experiment and simulation of droplet evaporation on the heated wall to examine the effects of a particle’s size and mass fraction and the wall’s temperature and wettability on the coffee ring effect. The experiment was carried out on a self-developed platform. The SPH simulation was operated on a GPU parallel algorithm. Compared with CPU computing, the acceleration ratio was reached more than 500 times for about 300,000 particles. The reliability of the algorithm is demonstrated by four benchmark cases.

Experimental results show that droplets with a high concentration of nano particles, when evaporated on the high-temperature wall, formed more uniform precipitates. The patterns are quite different from the morphology of coffee rings that was rendered by micro size particles in traditional studies. This article calls this precipitation pattern, one with a relatively uniform distribution, “coffee splat”. Coffee splat occurs when the wall temperature reaches 80 °C for a 100 nm particle with a mass fraction of 2.5 wt%.

Simulation with large rigid particles and small point–particles helps us to understand the effects of particle size and wall temperature on the deposition patterns. In the SPH simulation, we conducted dimensionless processing of data. It was found that the existence of large rigid particles significantly enhanced the formation of coffee rings, while small point–particles had a more uniform distribution, as compared to rigid particles. With the increase in the degree of wall superheat from 0.1 to 0.3, the number of deposited particles on the wall gradually decreased. Furthermore, this article has found that the spontaneous formation of large solid particles is related to the hydrophilicity of the wall’s surface. In summary, droplets with high concentration nanoparticles on a high temperature hydrophobic wall would form a uniform coffee splat, instead of a coffee ring. This may be a possible route to production of uniformly distributed and easily controlled nano-structure coatings by means of droplet evaporation.

## Figures and Tables

**Figure 1 nanomaterials-13-01609-f001:**
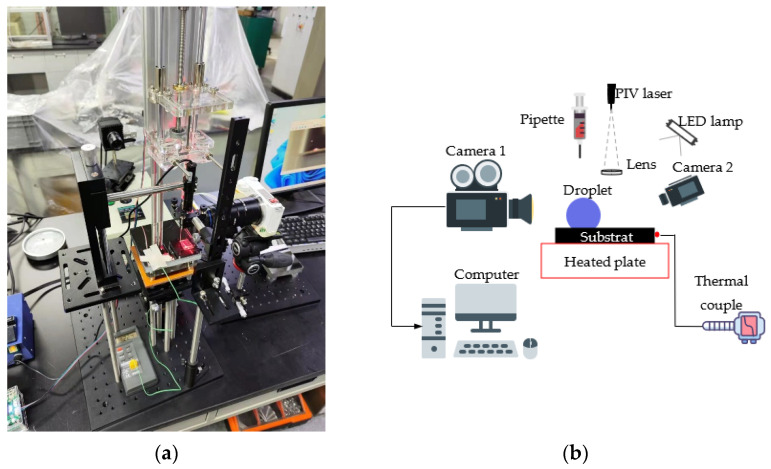
An (**a**) photo and (**b**) schematic diagram for the experimental setup.

**Figure 2 nanomaterials-13-01609-f002:**
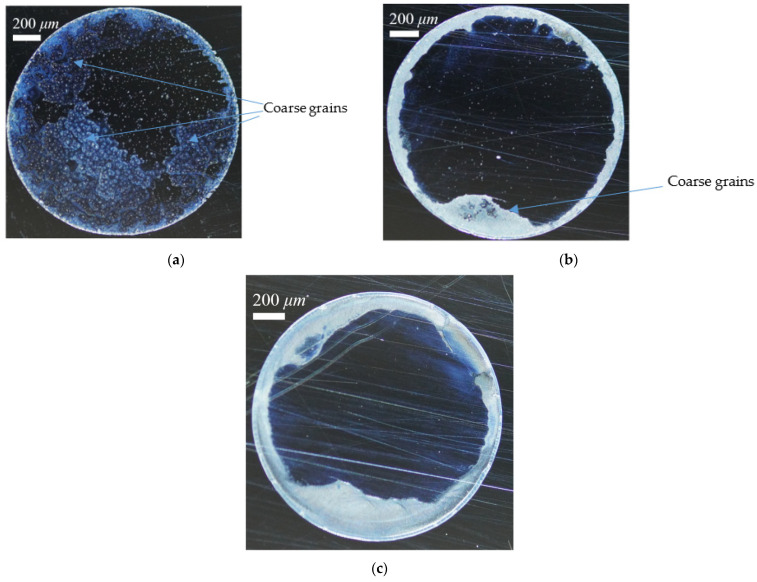
Particle deposition pattern on wall at a temperature of 40 °C, under different particle mass fractions: (**a**) 0.025 wt% of 10 μm, (**b**) 0.0125 wt% of 10 μm and 0.0125 wt% of 10 nm, and (**c**) 0.025 wt% of 10 nm.

**Figure 3 nanomaterials-13-01609-f003:**
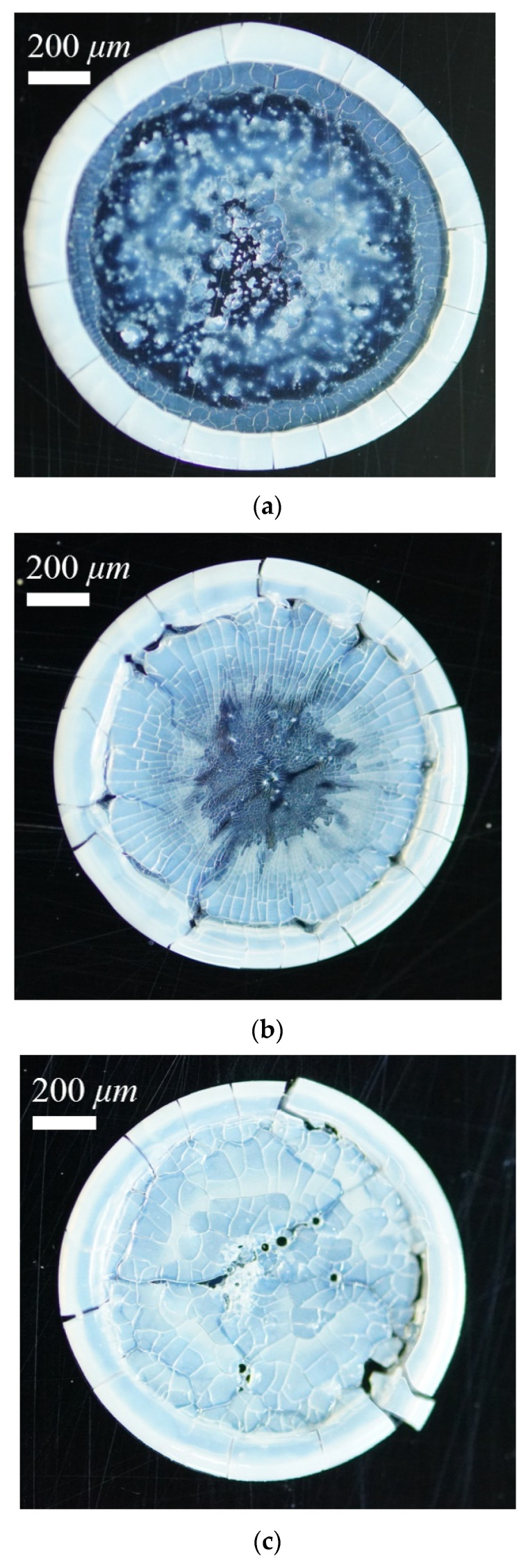
Particle deposition pattern, with nano particles only, with different wall temperatures: (**a**) 30 °C, (**b**) 50 °C, and (**c**) 80 °C.

**Figure 4 nanomaterials-13-01609-f004:**
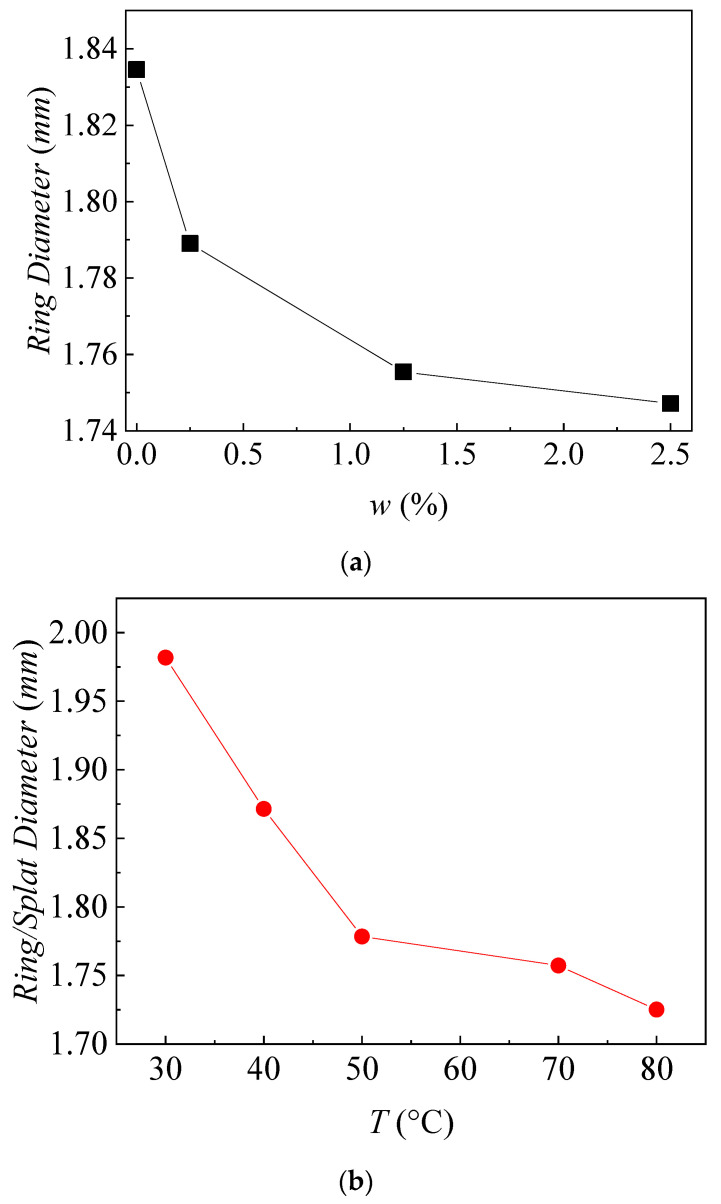
Diameter of coffee ring or splat with (**a**) different nano particle mass fraction for the cases in Figure 1, and (**b**) different wall temperature for the cases in Figure 2.

**Figure 5 nanomaterials-13-01609-f005:**
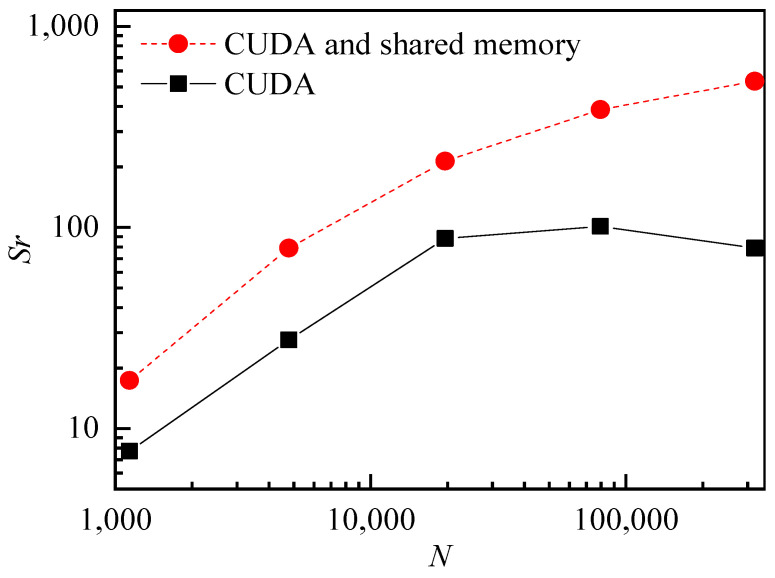
Speedup ratio (*Sr*) for different SPH particle numbers (*N*) using GPU (the black dot is the result of strategy one, and the red dot is the result of strategy two).

**Figure 6 nanomaterials-13-01609-f006:**
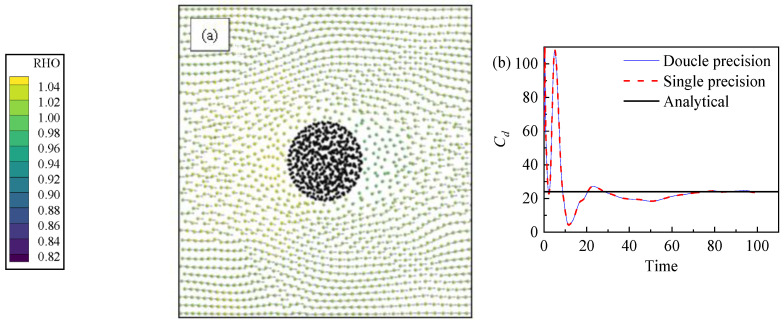
The (**a**) velocity distribution of the fluid in the presence of a single particle, and the (**b**) drag coefficient of the particle with time.

**Figure 7 nanomaterials-13-01609-f007:**
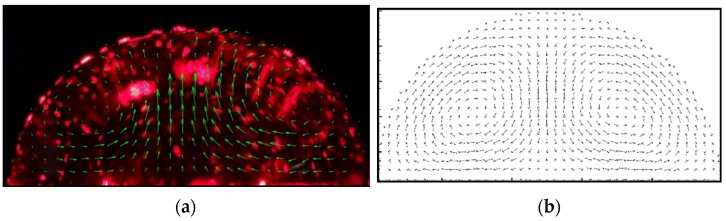
Droplet velocity distribution: (**a**) experimental result; (**b**) SPH numerical result. (Experimental boundary temperature is maintained between 72.5 °C to 73.5 °C, particle diameter is 10 μm, particle mass fraction is 0.025 wt% and numerical heated boundary is 75 °C); and (**c**) droplet center fluid vertical velocity along y direction.

**Figure 8 nanomaterials-13-01609-f008:**
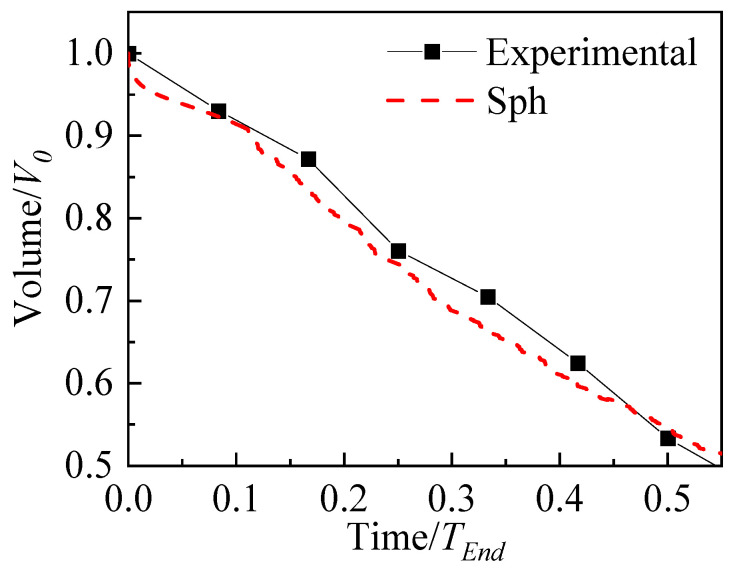
Droplet volume with time.

**Figure 9 nanomaterials-13-01609-f009:**
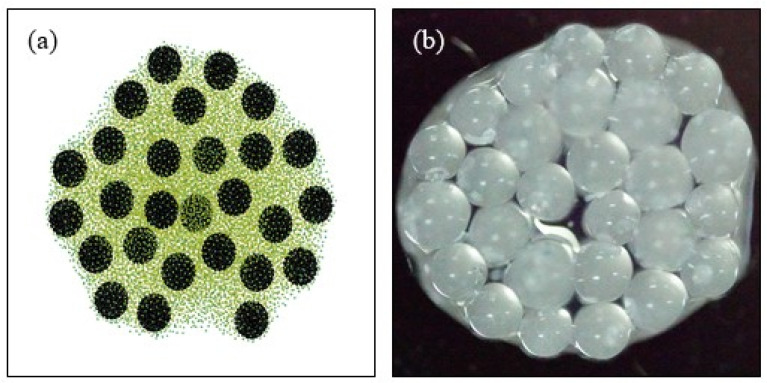
Particle deposition pattern: (**a**) numerical results with boundary temperature 56 °C and particles as set in experiments; (**b**) experimental photo of particle distribution after evaporation on wall temperature 50 °C, using 27 large particles with diameter 1000 μm, and small particles with diameter 100 nm in mass fraction 0.025 wt% and volume of 10 μL.

**Figure 10 nanomaterials-13-01609-f010:**
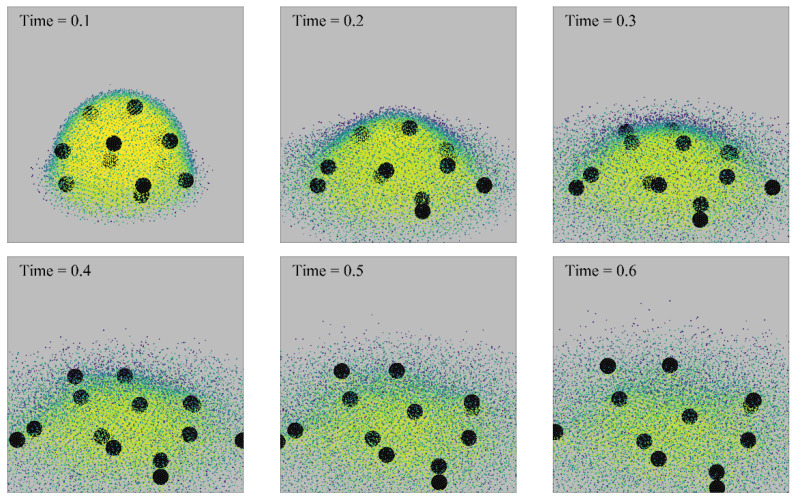
Morphology of a droplet evaporated on a wall (∆*T* = 0.1, *D_rigid_* = 0.25).

**Figure 11 nanomaterials-13-01609-f011:**
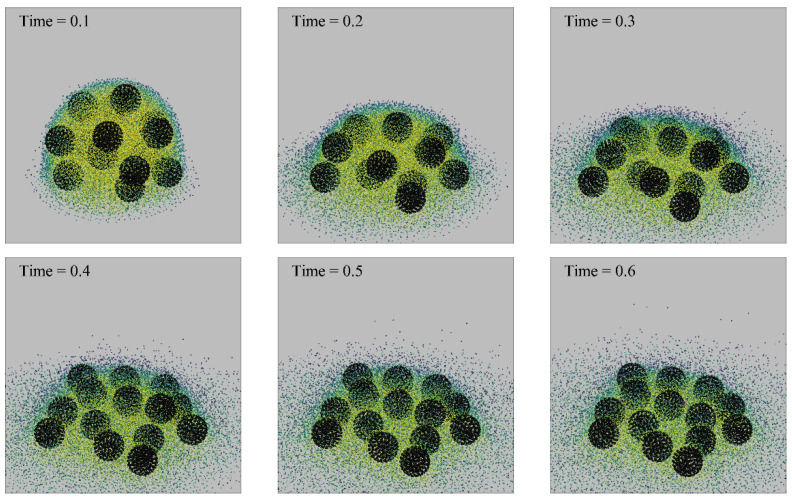
Morphology of a droplet evaporated on a wall (∆*T* = 0.1, *D_rigid_* = 1.0).

**Figure 12 nanomaterials-13-01609-f012:**
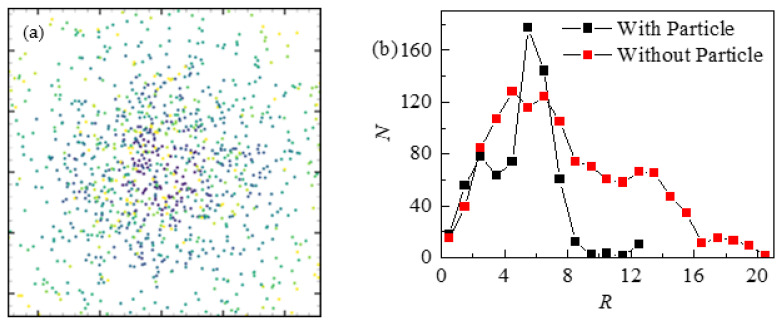
Distribution diagram of deposition point–particles: (**a**) the deposition result of particles after droplet evaporation in the absence of rigid particles, and (**b**) the radial distribution diagram of the number of deposited point–particles along the center of the droplet.

**Figure 13 nanomaterials-13-01609-f013:**
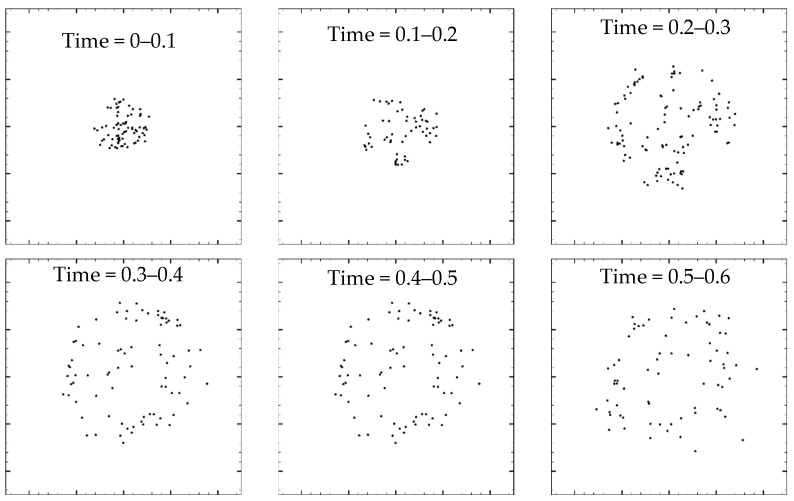
Deposition as particle distribution in different time ranges (Boundary temperature ∆*T* = 0.1).

**Figure 14 nanomaterials-13-01609-f014:**
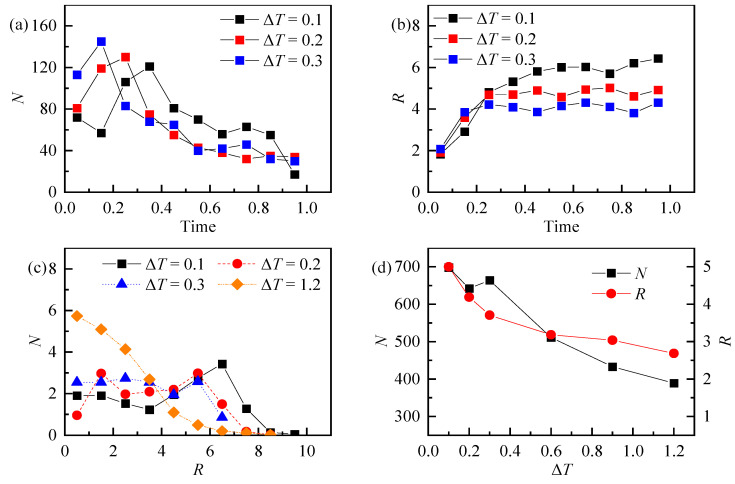
The distribution of deposited particles under different degrees of wall superheat: (**a**) the number of precipitated particles in different time periods, (**b**) the average distance between deposited particles and the droplet centroid in different time periods, (**c**) the average number of deposited particles in different distances from the center of mass, and (**d**) the relationship between the number of deposited particles and the average distribution range for different degrees of wall superheat.

**Figure 15 nanomaterials-13-01609-f015:**
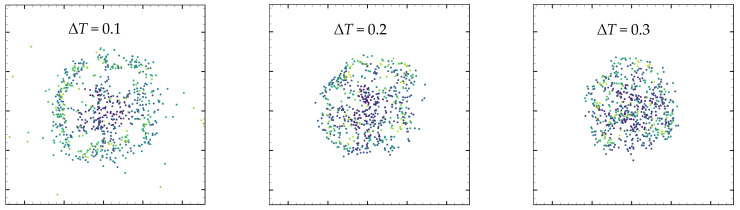
Point–particle distribution after droplet evaporation under different degrees of wall superheat (∆*T* = 0.1, 0.2, 0.3).

**Figure 16 nanomaterials-13-01609-f016:**
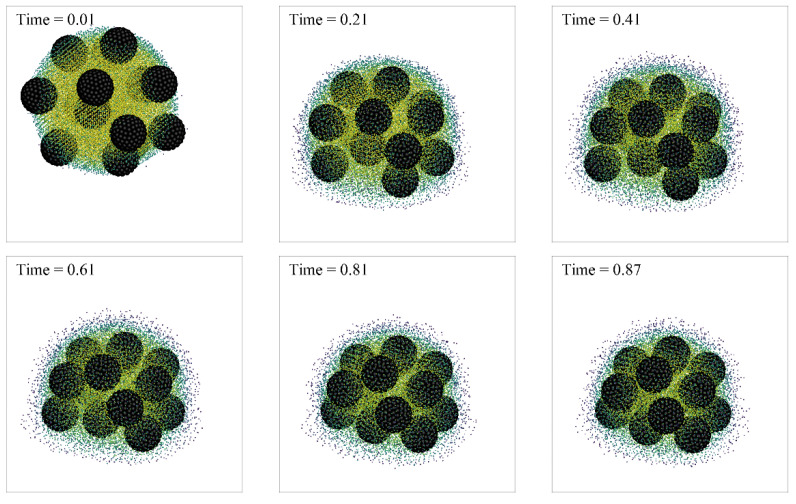
Regular double-tower arrangement of particles formed during evaporation on a hydrophobic wall.

**Figure 17 nanomaterials-13-01609-f017:**
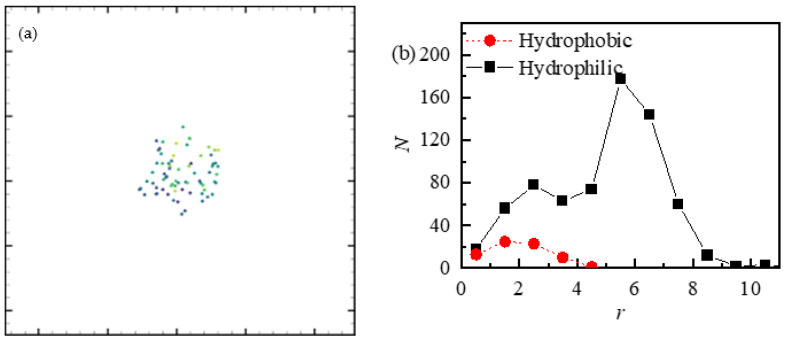
Distribution diagram of deposited point–particles on a hydrophobic wall: (**a**) the resulting deposition of particles, and (**b**) the radial distribution diagram of the number of deposited point–particles along the center of the droplet.

**Table 1 nanomaterials-13-01609-t001:** Dimensionless parameters and physical parameters of fluids.

Name	Symbol	Value	Units
Length scale	*L**	2.81 × 10^−9^	Meter
Time scale	*t**	1 × 10–9	Second
Mass scale	*M**	1.247 × 10^−23^	Kilogram
Temperature scale	*T**	562	Kelvin
Van der Waals equation constant	$\overset{-}{\alpha }$	74,379.1386	-
Van der Waals equation constant	$\overset{-}{\beta }$	0.583371015	-
Boltzmann constant	$\overset{-}{k}$	32,846.97509	-
Liquid shear Viscosity	*μ_l_*	101	-
Vapor shear Viscosity	*μ_v_*	10	-
Liquid thermal conductivity	*κ_l_*	10,425,020.05	-
Vapor thermal conductivity	*κ_v_*	104,250.2005	-

**Table 2 nanomaterials-13-01609-t002:** Simulation parameters for each subsection.

Section	Superheat Degree (Δ*T*)	Wettability
5.1	0.1, 0.2, 0.3	Hydrophilic
5.2	0.2, 1.2	Hydrophilic
5.3	0.1	Hydrophilic and Hydrophobic

## Data Availability

The data presented in this study are available on request from the corresponding author. The data are not publicly available because of privacy.

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
