# Peer review of "Modeling and Experiments of Droplet Evaporation with Micro or Nano Particles in Coffee Ring or Coffee Splat"

_nanomaterials, 2023, doi:10.3390/nano13101609_

Round 1

Reviewer 1 Report

Ti\his study solves two phase flow  in a droplet using the SPH method, and presents experimental data. I have major concerns with this study as follows:

(1) The SPH method should be validated first for one phase flow (no solid particles) by comparison with existing literature results or data

(2) The validation of the two phase flow in Figure (5) is not adequate. Qualitative results should be compared.

(3) The SPH method models a fluid continuum field with discrete fluid particles. What is the number of fluid particles in the SPH method compared with the number of the coffee solid particles. Should they be equal or not?

(4 ) AS the number of coffee particles increase, the volume of the solid particles should not exceed the volume of the droplet. It is shown in Figure (7b) that the volume of the solid particles is equal to the droplet volume. In this case there is no fluid in the droplet.

(5) In line 189- change " government equation" to governing equation.

Author Response

Dear Editors and Reviewers:

Thank you for your letter and for the reviewers’ comments concerning our manuscript. Those comments are all valuable, instructive and helpful improving our paper. We have read comments carefully and have made corrections. Revised portion are marked in the paper.

The main corrections in the paper and the responds to reviewer’s comments are point-to-point answered in the word file.

Reviewer 2 Report

The authors conducted experimental and numerical studies of droplet evaporation on a heated substrate in order to study the effect of particle size and mass fraction, substrate temperature and wettability on the coffee ring effect.

During the experiment, the authors released one drop when injected from a pipette onto a heated substrate with a constant temperature and recorded the process of its evaporation with two chambers. There have been several cases of experiments on evaporation of droplets without particles or with micro- and nanoparticles. The experiment used a particle image velocity meter and a high-resolution camera with a temperature-controlled heater and a computer for data collection. The results showed that nanoparticles can form a homogeneous coffee crumb instead of the usual coffee ring when using micro particles. To explain this phenomenon, the authors developed a complex multiphase model that included a fluid model with smooth particle hydrodynamics in combination with the van der Waals equation of state for droplet evaporation, a model of solid particles of microparticles of finite size and a model of point particles of nanometer particles. The results showed that as for solid particles, they spontaneously formed on the substrate, and their structure is mainly affected by the wettability of the boundary, but to a lesser extent by the fluid flow and thermal regime. Calculations show that at a low substrate temperature, it was easier for particles to settle on the contact line. At high substrate temperature, the effect of the coffee ring weakens, and the particles are more likely to settle in the center of the drop.

The authors concluded that their experimental and numerical results proved that particle size can play a significant role during particle deposition, which may be a possible way to obtain uniformly distributed coatings with nanostructures.

The work undoubtedly deserves to be published in the journal Nanomaterials.

Author Response

Dear Reviewer:

Thank you for your letter and for the reviewers’ comments concerning our manuscript. Those comments are all valuable, instructive and helpful improving our paper. We have read comments carefully and have made corrections. Revised portion are marked in the paper.

Thank you very much for your recognition of our work. Received your encouragement, we will continue to do in-depth research on micro and nano particle deposition in the coffee ring or coffee splat effects.

Reviewer 3 Report

Dear authors,

I read carefully your manuscript, named “Modeling and experiments of droplet evaporation with micro 2 or nano particles in coffee ring or coffee splat“.

You should definitely add some more specific data in Conclusion section of this paper. Also, there are many mistakes throughout the paper and these are my specific objections:

Line 39 – delete word and before droplet size and put comma instead;

Line 40 – unify the font on the end of page 1 and beginning of page 2;

Line 43 – you cannot use term we, paper cannot be written in plural first-person pronouns;

Line 46 – same as previous comment;

Line 48 – it is numerical, not numerically;

Line 54 – you cannot use term we, paper cannot be written in plural first-person pronouns;

Line 55 – same as previous comment;

Line 60 – same as previous comment;

Line 61 – where is section 0?;

Line 67 – you cannot use term we, paper cannot be written in plural first-person pronouns;

Line 75 – same as previous comment;

Line 77 – same as previous comment;

Line 100 – delete dot after word Table (Table. 1);

Line 101 – micro instead mirco;

Lines 114 and 115 – you cannot use term we, paper cannot be written in plural first-person pronouns;

Line 117 – you cannot use term our;

Line 136 – Table 1. – this is not a table, create real table;

Line 140 – Table 2. – this is not a table, create real table;

Line 159 - 182 – you need to write what every mark in equations stands for and write their units;

Line 189 – equation 10 is written twice;

Line 190 - 193 – you need to write what every mark in equations stands for and write their units;

Line 193 – equation 12 is missing;

Line 196 – write unit for a;

Line 204 – equation 18 instead equation 12;

Line 207 – you need to write what every mark in equations stands for and write their units;

Line 222 – Droplet instead droplet;

Line 224 - 229 – write units;

Line 233 – you cannot use term we, paper cannot be written in plural first-person pronouns;

Line 244 – memory instead memory inside Figure 3;

Line 244 – delete dot after word Table;

Line 260 – you cannot use term our;

Line 263 – delete dot after (3);

Line 280 - 282 – write units;

Line 304 – write a and b below or above figures;

Line 337 – you cannot use term we, paper cannot be written in plural first-person pronouns;

Line 416 – fill rest od page 17;

Best regards,

Reviewer

Author Response

Thank you for your letter and for the reviewers’ comments concerning our manuscript. Those comments are all valuable, instructive and helpful improving our paper. We have read comments carefully and have made corrections. Revised portion are marked in the paper.

The main corrections in the paper and the responds to reviewer’s comments are point-to-point answered in the enclosed word file.

Reviewer 4 Report

The present paper performed experimental and numerical studies on the coffee rings or coffee splats formed by droplet evaporation with micro or nano polystyrene sphere particles. To achieve this aim, Particle Image Velocimetry (PIV) and a high-resolution camera are used in the experiment, with a temperature-controlled heater and a data-acquisition computer. This reviewer examined that the present paper is well organized and well written, and the mathematical equations are well constructed which are supported by the references. However, for the readers' interest, a few comments or suggestions need to be incorporated by the authors raised by this reviewer.

 1.      The computational cost of your present numerical simulations needs to be discussed.

2.      Could you please provide the mathematical expressions of the boundary condition that you explained in the texts Section 3.1.1?

3.      In Figures 4(b), 6, and 10(b), could you please discuss the percentages of their differences?

4.      For clarity of the concluding remarks, please discuss the analysis of the results pointwise in the conclusion section.

5.      The reference list should be revised based on the Journal of Nanomaterials template.

Author Response

(The authors gave the same response as above.)

Round 2

Reviewer 3 Report

Dear authors,

You should add more specific data in Conclusion section of this paper. For example, when you write: With the increase of wall superheat degree, the number of deposited particles on the wall gradually decreases, mention how much this numer decreases, add samo values in conclusion. Also, there are other mistakes throughout the paper that you need to correct:

Line 42 – unify the font on the end of page 1 and beginning of page 2;

Line 45 – you cannot use term we, paper cannot be written in plural first-person pronouns;

Line 76 – 102 kPa instead 102kPa;

Line 98 – Clean lab desktop? Delete this, it is unnecessary;

Line 117 – you cannot use term we;

Line 120 – you cannot use term our;

Line 140 – put space between value and unit. Correct this on multiple examples in the paper;

Line 164 – instead: (unit: kg/m3), write: (kg/m3). Correct this for every unit added in the paper;

Line 165 – instead: (unit: m)is, write: (unit: m) is;

Line 170 – instead: ua(unit: J/kg), write: ua (unit: J/kg). You need to put space in between, correct this on multiple examples in the paper;

Line 260 – delete dot after word Table;

Line 262 – you cannot use term our;

Line 276 – you cannot use term our;

Line 349 – where is mark b) for right figure?;

Also, correct numbers of each section and subsection, all are written in the wrong way, no traceability. Each section is 1. up to page 16, and every subsection is 1.1. or 1.1.1. Correct this for whole paper.

Best regards,

Reviewer

Author Response

Dear Editors and Reviewers:

Thank you for your letter and the reviewer's comments on our manuscript. The second round of revision suggestions are very valuable for our paper. According to your suggestions, we have revised the paper according to your comments.

The main corrections in the paper and the response to reviewer’s comments are listed in the enclosed file.

Reviewer 4 Report

No more further comments.

Author Response

Dear Editors and Reviewers:

Thank you for your letter and the reviewer's comments on our manuscript. The second round of revision suggestions are very valuable for our paper. According to your suggestions, we have revised the paper according to your comments.

The main corrections in the paper and the responds to reviewer’s comments are as following:

Review 4:

No more further comments.

Answer:

Thank you very much for your comments.